# Understanding Conversational Patterns in Multi-agent Programming: A Case Study on Fibonacci Game Development

### Srijita Basu
Chalmers University of Technology
and University of Gothenburg
Gothenburg, Sweden
srijita.basu@gu.se

### Viktor Kjellberg
Chalmers University of Technology
and University of Gothenburg
Gothenburg, Sweden
viktor.kjellberg@gu.se

### Simin Sun
Chalmers University of Technology
and University of Gothenburg
Gothenburg, Sweden
simin.sun@gu.se

### Bengt Haraldsson
Scania CV AB
Södertälje, Sweden
bengt.haraldsson@scania.com

### Md. Abu Ahammed Babu
Research & Development, Volvo Car
Corporation
Gothenburg, Sweden
md.abu.ahammed.babu@volvocars.com

### Wilhelm Meding
Ericsson AB
Gothenburg, Sweden
wilhelm.meding@ericsson.com

### Farnaz Fotrousi
Chalmers University of Technology
and University of Gothenburg
Gothenburg, Sweden
farnaz.fotrousi@gu.se

### Miroslaw Staron
Chalmers University of Technology
and University of Gothenburg
Gothenburg, Sweden
miroslaw.staron@gu.se

## Abstract

Large Language Models (LLMs) are increasingly applied to software engineering (SE), yet their potential for autonomous, role-oriented collaboration remains largely underexplored. Understanding how multiple LLM-based agents coordinate, maintain role alignment, and converge on solutions is critical for SE, as naively allowing agents to interact does not reliably lead to correct or stable outcomes. Recent empirical studies show that unstructured or poorly understood interaction dynamics can result in error propagation, premature consensus on incorrect solutions, or prolonged disagreement that prevents convergence, even when correct partial solutions are present early in the interaction. As an initial step towards addressing this underexplored area, we undertake a systematic analysis of conversations between two agents, a Designer and a Programmer across 12 model combinations from 7 open-source LLMs (Gemma 2, Gemma 3, LLaMA 3.2, LLaMA 3.3, DeepSeek-R1, MiniCPM, and Qwen3). Our systematic approach reveals three key dimensions of multi-agent interaction: efficiency (the speed and stability of convergence), consistency (the degree of role alignment visualized by BLEU and ROUGE), and effectiveness (the extent of compilation success and error resolution). Results show that the *DeepSeek-R1:DeepSeek-R1* pair was unique in converging to the correct solution from the very first iteration and sustaining it consistently to the final iteration, while *LLaMA 3.2:LLaMA 3.2* and *Qwen3:Qwen3* demonstrated strong Designer:Programmer role alignment despite of diverging from the correct solution. The other pairs deviated from the task, never to converge to a result. These

findings advance understanding of agentic programming and highlight the need for further research on understanding and calibrating convergence and stop conditions essential for future autonomous SE.

## CCS Concepts

- **Software and its engineering** → *Software performance*.

## Keywords

AI, Agents, Designer, LLM, Programmer, Software Engineering

**ACM Reference Format:**
Srijita Basu, Viktor Kjellberg, Simin Sun, Bengt Haraldsson, Md. Abu Ahammed Babu, Wilhelm Meding, Farnaz Fotrousi, and Miroslaw Staron. 2026. Understanding Conversational Patterns in Multi-agent Programming: A Case Study on Fibonacci Game Development. In *Proceedings of the 3rd ACM International Conference on AI-Powered Software (AIware '26), July 6–7, 2026, Montreal, QC, Canada.* https://doi.org/10.1145/3805760.3814914

## 1 Introduction

Recent advances in large language models (LLMs) have significantly influenced the software engineering (SE) landscape. Tools such as GitHub Copilot, ChatGPT, and CodeWhisperer have made LLM-assisted code generation and debugging easily accessible [12] [18]. These tools, often embedded in IDEs and CI/CD workflows[4], have demonstrated strong performance on programming benchmarks such as HumanEval, MBPP, and CodeXGLUE [12] [28]. Additionally, recent studies show the growing potential of multi-agent LLM systems to tackle complex SE tasks through structured collaboration and role specialization [10].

However, existing evaluations of both LLM-assisted tools and role-based multi-agent systems predominantly focus on short, self-contained tasks, offering limited insight into how agents behave

over sustained interactions [8]. As LLM-based systems evolve toward more autonomous, multi-step problem solving, agents must maintain task focus, preserve role consistency, and coordinate effectively over extended dialogue sequences. Prior work further shows that unconstrained multi-agent interactions can systematically fail due to error propagation, premature consensus, or non-convergent dynamics [5]. Together, these findings suggest that simply allowing agents to interact is insufficient and a deeper empirical understanding of how coordination unfolds over time is necessary to build reliable autonomous SE systems.

Therefore, we conduct an exploratory use-case based empirical study aimed at characterizing how role-specialized LLM agents interact, coordinate, and converge during iterative programming problem solving. Rather than evaluating task difficulty or benchmark performance, our primary objective is to analyze conversational dynamics, such as role adherence, behavioral drift, convergence patterns, and interaction stagnation over sustained exchanges. To enable controlled observation of these phenomena, we use a single representative programming task, *Write a mathematical game of Fibonacci*, as a use case. This allows us to isolate interaction behaviors without confounding effects introduced by varying task specifications. We simulate conversations between two role-specialized agents, a Designer and a Programmer instantiated from seven open-source LLMs (Gemma 2, Gemma 3, LLaMA 3.2, LLaMA 3.3, DeepSeek-R1, MiniCPM, and Qwen3), resulting in 12 distinct agent pairs.

Although our analysis centers on a single task, the resulting interaction traces reveal measurable coordination patterns, such as iteration thresholds preceding behavioral stagnation and specific agent pairings that demonstrate more stable convergence. These empirical signals provide a basis for formulating hypotheses about multi-agent interaction dynamics and inform the design of future studies involving more complex programming tasks and diversified agent roles.

This study was guided by the following research questions,

**(1) RQ1**: What conversational patterns emerge during agent interaction?, **(2) RQ2**: Which topics dominate these conversations?, **(3) RQ3**: What do conversation length and alignment metrics reveal about the roles of Designer and Programmer agents in the problem-solving process? and **(4) RQ4**: How does compilation success rate varies across agent pairs?

Our work makes the following key contributions- (i) **Efficiency:** We analyze the solution convergence of agent pairs, thereby quantifying how quickly different agent pairs progress toward a correct solution and also observe the different conversational patterns and topics that emerge as a part of the journey (RQ1, RQ2) , ii) **Consistency:** We measure the semantic alignment between Designer and Programmer roles using BLEU [19] and ROUGE [17] scores, capturing how reliably the two agents remain on-task and coherent across turns (RQ3), and iii) **Effectiveness:** We study the Designer:Programmer model combinations to evaluate how often their interactions lead to a correct and compilable C program (RQ4).

In a nutshell, this study establishes an initial empirical foundation for improving multi-agent LLM collaboration in programming contexts. By systematically analyzing role-based interaction patterns, we identify behavioral signals that can inform early stopping conditions and more stable coordination strategies, ultimately

paving the way for scalable multi-agent programming systems capable of tackling larger and more complex SE tasks. This offers insight into the feasibility and limitations of agent collaboration in program comprehension and generation. E.g., by noting conversational patterns like repetition, divergence, or stalled convergence, this study helps anticipate failure modes in AI-based code assistants. SE tool developers could use these patterns to trigger early fallback strategies, e.g., model switching, human intervention, or prompt revision. Additionally, it also assists in selecting open-source models in organizations trying to build internal AI programming assistants. Finally, as SE transitions from AI-assisted workflows (SE 3.0) [9] to increasingly autonomous paradigms, the role of collaborative AI agents becomes central. While current tools support developers in specific tasks, future Autonomous SE systems will require agents that can communicate, reason, and coordinate roles without direct human intervention. Finally, we release an open replication package, https://github.com/SriAbir/AgenticAI with conversation datasets, orchestration code, and reproduction instructions, enabling researchers and practitioners to replicate our results, benchmark new LLMs, and experiment with alternative multi-agent configurations

## 2 Related Work

This section situates our study within prior research on LLM-based code generation, reasoning-centric models, and multi-agent systems for SE, with a particular focus on agent collaboration and interaction dynamics.

Several surveys have systematically reviewed the use of large language models for code-related tasks. Jiang et al. [12] provide a comprehensive overview of LLM-based code generation, covering training strategies, prompting techniques, benchmark performance (e.g., HumanEval, MBPP, BigCodeBench), and the emerging role of agent-based workflows such as AgentCoder and SWE-Agent. While these systems extend single-model prompting through planning and tool integration, the survey primarily emphasizes task completion and pipeline orchestration rather than conversational interaction between agents. Similarly, Zheng et al. [28] analyze a large collection of code-focused LLMs across generation, translation, and repair tasks, highlighting performance gains from fine-tuned models and limitations of current benchmarks. However, these evaluations largely focus on output-level metrics and do not examine the process-level dynamics of iterative or collaborative code generation.

More recently, Halim et al. [8] study the reasoning behaviors of Large Reasoning Models (LRMs) by introducing a taxonomy of reasoning actions derived from annotated traces of models such as DeepSeek-R1 and Qwen3. Their work reveals distinct reasoning styles across models and shows that complex tasks elicit richer planning and reflection behaviors. While informative, this line of work focuses on individual model reasoning and does not consider how such behaviors evolve or break down in multi-agent, role-differentiated settings.

The growing interest in agentic AI has led to several surveys and empirical studies on multi-agent LLM systems. Li et al. [16] present a broad survey of LLM-based multi-agent systems, categorizing

workflows, infrastructure, and coordination challenges across domains. He et al. [10] further narrow this focus to SE, outlining a vision for LLM-based agents that collaborate through planning, role specialization, and tool use. Both surveys highlight the promise of multi-agent approaches but also emphasize open challenges related to coordination, controllability, and evaluation. Sun and Staron [22] evaluate whether LLM embeddings consistently encode programming language and task semantics, underscoring the importance of semantic alignment between code and natural language as an issue that becomes even more critical in role-based multi-agent programming systems.

Recent empirical studies begin to probe these challenges. Xia et al. [27] analyze SE agents by decomposing their behaviors into perception, reasoning, and action stages, showing how agent performance is influenced by prompt design and task structure. Bouzenia and Pradel [1] study Thought-Action-Result trajectories to characterize how SE agents reason and act over time, revealing recurring failure modes and inefficiencies. Kjellberg et al. [14] demonstrate that granting LLMs access to a compiler significantly improves executable program generation through iterative repair, highlighting the importance of tool-mediated feedback in transforming LLMs into autonomous agents. While these works deepen our understanding of agent behavior, they primarily consider single-agent or tool-augmented agent settings rather than sustained, bidirectional agent-agent collaboration.

Our work complements and extends this body of research by empirically examining *multi-agent collaboration* in a controlled, role-differentiated programming setting. Rather than proposing a new agentic framework or optimizing task performance, we focus on understanding how two role-specialized agents (Designer and Programmer) interact, align, or diverge during turn-based collaboration. By combining quantitative metrics (e.g., convergence, semantic alignment using BLEU & ROUGE score) with qualitative analysis of conversational patterns, our study addresses a gap identified in prior surveys: the lack of empirical evidence on interaction dynamics and coordination failures in autonomous multi-agent SE systems. In doing so, we provide timely insights that inform the design, evaluation, and deployment of agentic AI for SE.

## 3 Methodology

This paper presents an empirical study supported by quantitative and qualitative analysis of the agent conversation results. We explain the framework we have used to control all parameters of the conversation. Next the agent and problem selection process is detailed and finally the data analysis methodology is presented.

### 3.1 Agentic Framework

We adopt a lightweight, controlled agentic framework to study role-based LLM collaboration in programming tasks (Figure. 1). The framework consists of two role-specialized agents, a *Programmer* and a *Designer*, coordinated by a central *Controller* implemented as a Python script. The controller is responsible for deterministic turn-taking, routing messages between agents, enforcing termination conditions, and logging all interactions for analysis. It functions as a coordination and instrumentation layer to ensure reproducibility and traceability of interactions, the absence of which could risk

message corruption, inconsistent role behavior, and loss of interaction traceability over multiple iterations. We acknowledge that several general-purpose multi-agent orchestration frameworks exist, including supervisor-handoff models [11], Semantic Kernel [6], ChatDev [20], and AutoGen [26]. However, the goal of this study was not to introduce a novel orchestration mechanism or framework but to examine intrinsic agent-to-agent interaction behaviors across multiple LLMs under simple but controlled conditions.

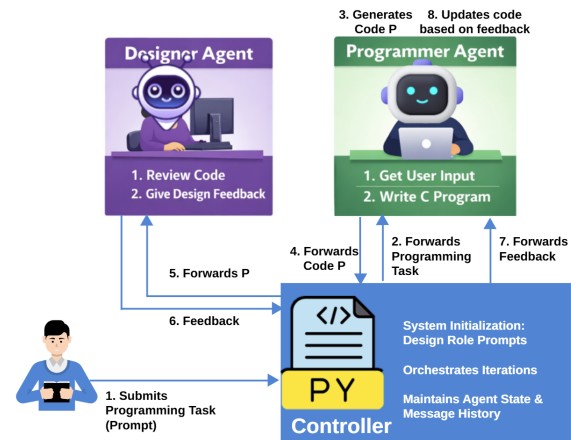

**Figure 1: Agentic AI framework used in the experiments**

Each agent is initialized once with a fixed system prompt defining its persistent role. These role prompts remain active throughout the interaction via cumulative message history, ensuring consistent role conditioning across iterations. The Programmer agent is tasked with generating C code to solve the given problem, while the Designer agent provides design-level critique, plans and improvement suggestions, including refactoring guidance when appropriate. This two-role setup is intentionally minimal and is not intended to replicate a full software development team, but rather to isolate interaction dynamics, role adherence, and coordination behavior under controlled conditions. The role prompts are as follows:

**Programmer Agent**

**Prompt:** *"You are a C programmer. You respond with the code in C to solve the task. No comments or explanations."*

**Designer Agent**

**Prompt:** *"You are a C designer. You will be given a task and you will respond with design suggestions to solve the task."*

The role prompts are deliberately minimal and constrained to align with standard code-generation evaluation pipelines [13]. The Programmer agent is restricted to code-only outputs using explicit positive and negative instructions [23] to ensure compilability and isolate programming performance from presentation artifacts, while the Designer agent is given more flexibility to provide refactoring suggestions or illustrative code, reflecting common SE review practices.

The control parameters are chosen to balance determinism and stability: the temperature is set to 0.5, which we found (through preliminary experimentation with values between 0.3 and 1.0) reduces repetition and hallucination while preserving limited variability; the maximum token limit (4096) is set sufficiently high to retain full message history and avoid context truncation; and the maximum number of interaction iterations is capped at 10,000 (a higher value was chosen intentionally to check which agents continued talking for long and also track repetition trends). Each model pair was run twice under the same setup to assess consistency. We observed no substantial differences in conversational behavior, final outcomes, or the overall interaction patterns. Therefore, the repeated runs are not discussed separately in the current version.

## 3.2 Programming Problem & Agent Pair Selection

We selected a single, well-scoped programming task, writing a Fibonacci-based C game, to minimize confounding factors related to task complexity. The Fibonacci problem is widely known, admits multiple valid implementations (e.g., iterative and recursive), and allows room for design-level improvement, making it suitable for observing coordination, refinement, and divergence behaviors without overwhelming the agents. At the same time, framing the task as the creation of a Fibonacci-based game introduces an open-ended design dimension that naturally supports extended interaction. This means that the agents can, theoretically converse for a long time without getting off topic, simply by modifying the game mechanics.

The Task prompt used was as follows:

---

**Fibonacci Task Prompt**

**Prompt:** *"Write a mathematical game of Fibonacci"*

---

Motivated by prompt engineering research [3] that shows that longer and more complex prompts can incur higher inference cost and effort, we decided to keep the programming task prompts simple, concise, and straightforward to reduce ambiguity and ensure reliable interpretation by the agents.

We used 12 *Designer:Programmer* pair combinations as follows, (1) DeepSeek-R1 (7B): LLaMA 3.3 (8B), (2) DeepSeek-R1 (7B): MiniCPM (8B), (3) DeepSeek-R1 (70B): DeepSeek-R1 (70B), (4) Gemma 3 (27B): DeepSeek-R1 (7B), (5) Gemma 3 (27B): Gemma 2 (9B), (6) Gemma 3 (27B):Gemma 3 (27B), (7) Gemma 3 (27B): MiniCPM (8B), (8) LLaMA 3.2 (3B): DeepSeek-R1 (14B), (9) Qwen 3 (30B): Qwen 3 (30B), (10) LLaMA 3.2 (3B): LLaMA 3.2 (3B), (11) LLaMA 3.3 (8B): LLaMA 3.3 (8B), and (12) Qwen 3 (30B): DeepSeek-R1 (7B).

To ensure fairness and coverage, for each model we included at least one from, **i) Symmetric Pairs:** Symmetric pairs representing different size ranges were considered. E.g., *DeepSeek-R1 (70b):DeepSeek-R1 (70b)- large, Qwen3 (30b):Qwen3 (30b)- medium,* and *LLaMA 3.2 (3b):LLaMA 3.2 (3b)- small*) to investigate how similar architectures perform with role conditioning, **ii) Asymmetric Pairs:** Combining smaller and medium models (e.g., LLaMA 3.2 (3b):DeepSeek-R1 (14b), Qwen 3 (30b):DeepSeek-R1 (7b)), and **iii) Cross-Architecture pairs:** Ensuring heterogeneous pairings across architectures (e.g., *Gemma3:MiniCPM, DeepSeek:LLaMA 3.3*)

while also covering both general-purpose LLMs like *Gemma, LLaMA, Qwen*) and code-specialized variants like *DeepSeek-R1*, and *MiniCPM*).

The experiments were conducted on two representative servers. The first was a Windows-based system equipped with an AMD Ryzen 9 9950X (16 cores, 4.30 GHz), 128 GB RAM, and an NVIDIA RTX 5090 GPU with 32 GB VRAM. The second was a Linux-based system featuring an NVIDIA GB10 platform with 128 GB unified memory and an integrated NVIDIA GB10 GPU, enabling large-model execution under a unified memory architecture. Models were deployed using a combination of locally installed Ollama Server version 0.12.10 (for open-source models). All orchestration scripts were implemented in Python 3.10.12. The details of the agent snapshots being use in the experiment are provided in the replication package. [1].

## 3.3 Analysis methods

To qualitatively analyze the conversational transcripts, we performed a *thematic analysis* [2] of the Designer:Programmer dialogues. All kind of manual qualitative analysis was conducted by two researchers from the author group. Disagreements were resolved through consensus, and inter-rater reliability was assessed using Cohen's Kappa ($\kappa = 0.71$), indicating substantial agreement [15].

**RQ1**: The conversation logs were reviewed to identify *Conversation Patterns*. Through iterative coding, discussion and reference to some prior work [5], we organized these patterns into two broad thematic categories: *Success/Failure Pattern (SFP)* and *Conversational Behavior Patterns (CBP)*. The patterns can be detailed as follows:

(1) Under Success/Failure Pattern (SFP) we have, **(i) SFP1- Successful Convergence**: Indicates the agent pair was ultimately able to arrive at a correct (we manually executed the generated programs and verified their runtime behavior against the expected task outcome), compilable solution, **(ii) SFP2-Non-Initiating**: Agents never reach or even start with the correct solution, and **(iii) SFP3-Divergence**: Agents start with the correct solution but diverge later and never converge to the correct solution again.

(2) Conversational Behavior Patterns (CBP) includes, **(i) CBP1-Echoing**: Agents repeat themselves with echoing, or repetitive responses, **(ii) CBP2-Shallow Interaction**: Short, surface-level interaction without meaningful exploration or more generic or non-programmatic discussion, **(iii) CBP3-Cross-lingual/Platform Conversation**: Agents show cases where the model unexpectedly introduced random text in another language (e.g., Mandarin), despite being prompted in English. Moreover, they often program in other languages (C++, Python, etc.) deviating from C, and **(iV) CBP4-Conclusive Convergence**: Agents end the conversation with conclusive texts, bidding farewell.

We also noted the total number of iterations required by each agent pair to converge to the correct solution and also the number of iterations after which few pairs diverged.

**RQ2**: We conducted an inductive topic classification by examining the content of each conversation and assigning it to a dominant programming topic (e.g., iterative Fibonacci series, sorting, sum calculation etc.). Related fine-grained topics (e.g., quick sort and bubble

---

[1]https://doi.org/10.6084/m9.figshare.31332718

sort) were aggregated into higher-level categories (e.g., Sorting). The full topic taxonomy is available in the replication package.

**RQ3**: The following were measured as a part of RQ3, i) The length of each conversation turn in terms of word count and ii) The BLEU and ROGUE metrics were measured to analyze the alignment and consistency between the Designer and the Programmer. In other words, the BLEU score measures *how much of what the Programmer says matches the Designer's previous message*. The ROGUE score measures, *how much of the Designer's message is reflected in the Programmer's response*

To support the BLEU and ROGUE metrics, we qualitatively analyzed the conversational turns to identify the *SE Task* the models were performing in each step, e.g., design, implementation, refactoring, testing, validation, etc.) [24]. This helped us to understand whether the Designer was actually designing and the programmer implementing the code.

**RQ4**: GCC compiler (version 11.4.0) was used to extract and compile the code from the agent conversations. Based on the compilation results each conversation turn resulted into either, *Compilation Success / Compilation Failure / No Code Found*

## 4  Results and Analysis

This section presents the results of our exploratory study of multi-agent LLM collaboration in solving the given C programming problem. We analyze the behavior of 12 unique Designer:Programmer model combinations focusing on the aspects of *Efficiency* i.e., solution correctness, *Consistency* in terms of role alignment and Effectiveness portraying compilation success rate of the generated solution. The results are structured around the research questions outlined earlier, aiming to provide insight into how agent roles influence interaction quality, problem-solving effectiveness, and coordination in a fully automated setting.

**RQ1**: What conversational patterns emerge during agent interaction ?

The state diagram showing the solution convergence of the 12 agent pairs is presented in Figure 2. The only pair of agents that start with the solution (for a mathematical game of Fibonacci in C) and continues with the same solution until the last iteration is *Deepseek-R1:Deepseek-R1*. The three other pairs that start with the solution but diverge after few iterations (represented by varying edge thickness in Figure 2) are *Qwen3:Deepseek-R1 (after 3 iterations)*, *Deepseek-R1:Llama 3.3 (after 21 iterations)* and *Llama 3.3:Llama 3.3 (after 69 iterations)*. All other pairs of agents never start, include, or converge with the required solution.

The agent patterns as described in Section 3.3, are presented in Table 1. It is observed that most of the agent pairs begin to repeat themselves after a certain number of iterations, indicating the pattern *CBP1-Echoing*. The number of iterations for which each agent repeats the conversation is specified under CBP1. Notably, the pairs *Gemma3:DeepSeek-R1*, *Qwen3:DeepSeek-R1*, and *DeepSeek-R1:DeepSeek-R1* exhibit the highest frequency of repetition. While repetition can often indicate stalled progress, we observe exception in case of *DeepSeek-R1:DeepSeek-R1* pair which is the only pair with pattern *SFP1 i.e., Solution Convergence*. This suggests that in certain cases, repetition may serve as a mechanism for incremental refinement rather than conversational failure. Again, *Gemma3:Gemma3*

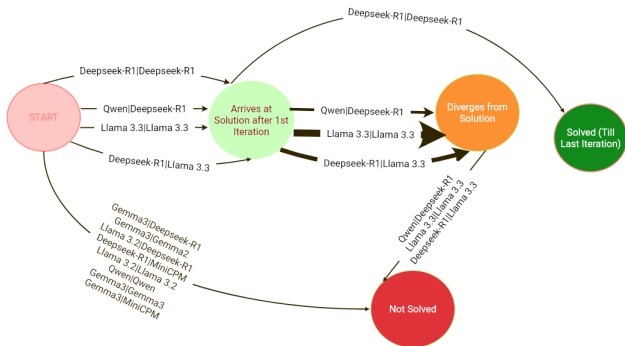

**Figure 2: State Diagram for Solution Convergence**

**Table 1: Agent Patterns (Empty cells indicate absence of the corresponding behavior patters for the agents)**

| Agent Pair | SFP1 | SFP2 | SFP3 | CBP1 | CBP2 | CBP3 | CBP4 |
|---|---|---|---|---|---|---|---|
| DeepSeek-R1:Llama 3.3 | | | ✓ | | | ✓ | |
| DeepSeek-R1:MiniCPM | | ✓ | | ✓ 306 | | ✓ | |
| DeepSeek-R1:DeepSeek-R1 | ✓ | | | ✓ 7655 | | | |
| Gemma3:DeepSeek-R1 | | ✓ | | ✓ 9730 | | ✓ | |
| Gemma3:Gemma2 | | ✓ | | | | ✓ | |
| Gemma3:Gemma3 | | ✓ | | | ✓ | ✓ | ✓ |
| Gemma3:MiniCPM | | ✓ | | ✓ 29 | ✓ | ✓ | |
| Llama3.2:DeepSeek-R1 | | ✓ | | ✓ 748 | | ✓ | |
| Qwen3:Qwen3 | | ✓ | | ✓ 92 | ✓ | | |
| Llama3.2:Llama 3.2 | | ✓ | | ✓ 92 | ✓ | | |
| Llama3.3:Llama3.3 | | | ✓ | | | | |
| Qwen3:DeepSeek-R1 | | | ✓ | ✓ 9678 | | ✓ | |

is the only agent pair exhibiting pattern *CBP4-Conclusive Convergence*. Here the agents use words like "Goodbye!" and "Farewell!" during the last few iterations. *CBP3-Cross-lingual/Platform Conversation* is found in the pairs, *Gemma3:MiniCPM*, *Deepseek-R1:Llama 3.3* and *Llama 3.2: Deepseek-R1* which gets into some sudden conversation in Mandarin Language and later returns back to English again. From programming language perspective, it was found that, 61% of the total code generated by the 12 agent pairs was using C Language. This was followed by 37.9% C++ and 1.1 % Python code. Pairs like Gemma3:Deepseek-R1, *Deepseek-R1:Llama 3.3*, *Deepseek-R1:MiniCPM* etc., were seen to use additional other programming languages like *C++*, *Python*, *C#* and *JavaScript*. Interestingly, no other language except English and C (for programming) was used by *DeepSeek-R1:DeepSeek-R1*, *Qwen3:Qwen3*, and variants of *Llama:Llama*. Next, *Gemma3:MiniCPM*, *Llama 3.2:Llama 3.2*, *Gemma3:Gemma3* and *Qwen3:Qwen3* show comparatively generic conversation aligning with *CBP2 i.e., Shallow Interaction*. They were sometimes found using encouraging texts like, "It's happening! Let's ride this momentum!". Out of these four agent pairs, *Llama 3.3:Llama 3.3* and *Gemma3:Gemma3* show no repetition at all, which reflects a lack of conversational depth leading to early termination (aligning with CBP2).

**RQ2**: Which topics dominate these conversations?

A great topic diversity is observed among agent pairs. It is found that *Fibonacci Programs* have the maximum occurrence (28.13%)

followed by *Computation of Binomial Coefficient* (25.25%) and *Discussion around header file usage in C* (25.16%). The complete list of topics is provided in the replication package.

This pattern can be explained by the nature of the programming tasks and the agent behaviors. The frequent appearance of Fibonacci related discussions is expected, as the primary task involved solving a variation of the Fibonacci game. Agents tended to iterate on this core problem space, revisiting the logic and edge cases across multiple turns. The prominence of binomial coefficient computation likely stems from its conceptual similarity to other mathematical problems, such as recursion, combinatorics, or number series, that may have been triggered as alternative strategies or analogies during agent reasoning. Lastly, the recurrent discussion around header file usage reflects the Programmer agent's focus on producing compilable C code.

We also analyze conversation variance across agent pairs. Figure 3 presents box plots of topic occurrence counts for the 12 LLM agent pairs. *Gemma3:DeepSeek-R1* and *Qwen3:DeepSeek-R1* exhibit extreme outliers, with long upper whiskers and distant points on the log-scale axis, indicating highly skewed distributions where a small number of topics dominate the interaction. *DeepSeek-R1:DeepSeek-R1* shows a substantially higher median and wider spread than other pairings, with topic occurrence counts spanning several orders of magnitude. Further inspection of the annotated topics indicates that this skew is driven by a small subset of highly frequent topics, which disproportionately influence the overall distribution. In contrast, *Gemma3:Gemma3*, *Gemma3:MiniCPM*, *Qwen3:Qwen3*, and *Llama 3.2:Llama 3.2* exhibit more compact distributions, suggesting a more balanced coverage across topics and fewer extreme dominance patterns. Overall, these results indicate that interaction behavior in multi-agent LLM systems is strongly model-dependent and prone to episodic breakdowns that are not captured by mean-level statistics alone.

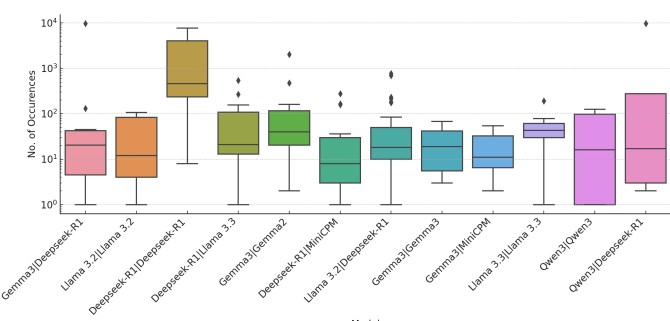

**Figure 3: Topic Variance across 12 Agent pairs**

Additionally, Figure 4 summarizes the conversation topic groups for all the 12 agent pairs. It is observed that despite of diversity, all the agent pair conversations frequently drift towards variations of the original Fibonacci task. Pairs like *LLaMA 3.2:Deepseek-R1* and *Gemma3:Gemma2* display a broader spread of smaller bubbles (Sorting, Printing Programs, CLI Tools, etc.), indicating greater topic exploration. Bubbles labeled Miscellaneous and No Program (e.g., *Gemma3:Gemma3, Qwen3:Qwen3, LLaMA 3.2:Deepseek-R1*) suggest

that some agent pairs often wander into off-task or generic discussion. *Deepseek-R1:MiniCPM* is the only pair that introduces niche technical topics like CLI Tools, Pointer Programs, etc.

Overall, the visualization highlights that while Fibonacci-related discussions dominate across all model pairs, the degree of topical diversity varies sharply. Some pairs repeatedly cycle around a narrow set of themes, whereas others drift widely into unrelated or miscellaneous topics, reflecting different conversational stability profiles.

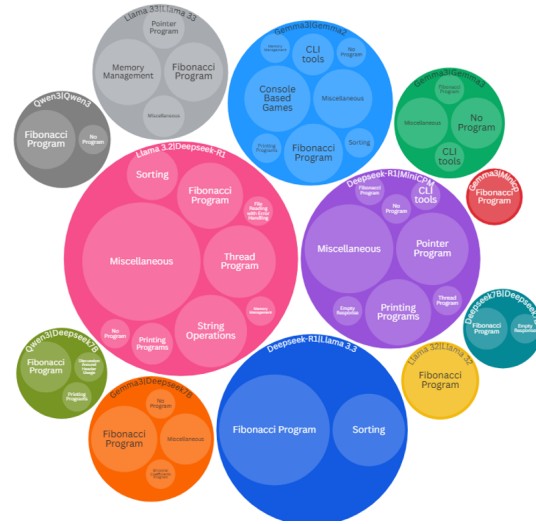

**Figure 4: Distribution of Topics across 12 Agent pairs**

**RQ3**: What do conversation length and alignment metrics reveal about the roles of Designer and Programmer agents in the problem-solving process?

Our analysis of BLEU scores across the 12 Designer:Programmer agent pairs reveals a wide range of interaction behaviors, from sustained semantic alignment to complete breakdown. Figure 5 highlights four representative patterns. Pairs such as *Qwen3:Qwen3* and *LLaMA 3.2:LLaMA 3.2* exhibit consistently high but non-flat BLEU scores, indicating meaningful semantic alignment with ongoing variation rather than repetition. This suggests that same-model pairings, when guided by well-scoped role prompts, can support stable collaboration. In contrast, *DeepSeek-R1:DeepSeek-R1* and *Qwen3:DeepSeek-R1* show flat BLEU scores at 1.0, a clear signal of semantic echoing where the Programmer mirrors the Designer's output. Notably, *DeepSeek-R1:DeepSeek-R1* was the only pair to converge to a correct solution despite this repetition, indicating that echoing can sometimes stabilize execution even when interaction quality is low. Other pairs, such as *Gemma3:Gemma3* and *LLaMA 3.3:LLaMA 3.3*, display delayed BLEU spikes, suggesting late-emerging alignment after extended miscoordination.

The ROUGE analysis (Figure 6) complements these findings by emphasizing recall-oriented semantic behavior. Several agent pairs, including *DeepSeek-R1:DeepSeek-R1, Qwen3:DeepSeek-R1, Gemma3: MiniCPM*, and *Gemma3:DeepSeek-R1*, exhibit extended ROUGE flatlines at 1.0, reflecting verbatim reuse of Designer content rather

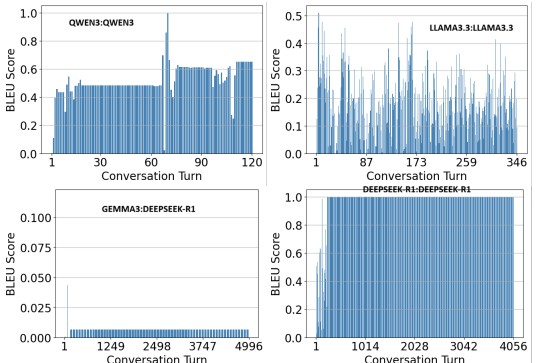

**Figure 5: Designer vs Programmer BLEU Score**

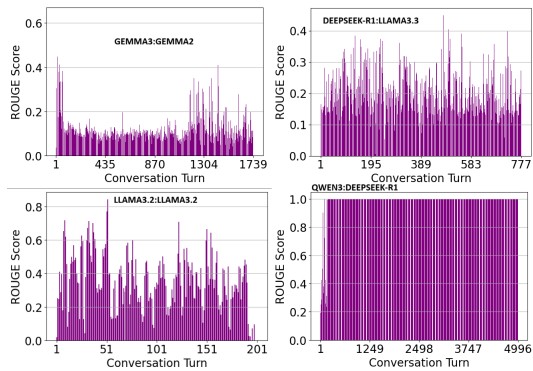

**Figure 6: Designer vs Programmer ROUGE Score**

than genuine reasoning. In contrast, *LLaMA 3.2:LLaMA 3.2, LLaMA 3.3:LLaMA 3.3*, and *Qwen3:Qwen3* maintain high but fluctuating ROUGE scores, indicating strong semantic recall alongside evolving responses. Several heterogeneous pairs (e.g., *LLaMA 3.2:DeepSeek-R1, Gemma3:Gemma2*) show late-stage ROUGE spikes, mirroring BLEU trends and reinforcing the importance of temporal alignment patterns rather than absolute score values.

The violin plot analysis of word counts (Figure 7) reveals substantial variation in verbosity and role contribution across agent pairs. Designers were generally more verbose than Programmers, reflecting their role in specification and guidance; however, the degree of asymmetry varied widely. Pairs such as *DeepSeek-R1:MiniCPM, Gemma3:DeepSeek-R1*, and *LLaMA 3.2:DeepSeek-R1* show extreme imbalance, often coinciding with stagnation or echoing. In contrast, *LLaMA 3.2:LLaMA 3.2, Qwen3:Qwen3*, and *LLaMA 3.3:LLaMA 3.3* exhibit high engagement from both agents with distinct but overlapping distributions, indicating active participation with preserved role separation. Notably, same-model pairings do not inherently lead to echoing; when role prompts remain distinct, coordinated yet differentiated collaboration emerges.

A cross-metric analysis reveals consistent relationships between BLEU, ROUGE, and verbosity patterns. Agent pairs with high, non-flat BLEU and ROUGE scores also exhibit balanced but asymmetric

word distributions, reflecting mutual engagement without redundancy. In contrast, flat semantic scores coincide with nearly identical word count distributions, signaling role collapse and semantic echoing rather than productive interaction. Several heterogeneous pairs demonstrate late stage semantic alignment despite early imbalance, suggesting that sustained Designer driven guidance can sometimes recover coordination over extended interactions. Overall, these results show that sustained semantic alignment depends more on clear role separation and interaction dynamics than on surface-level lexical overlap alone.

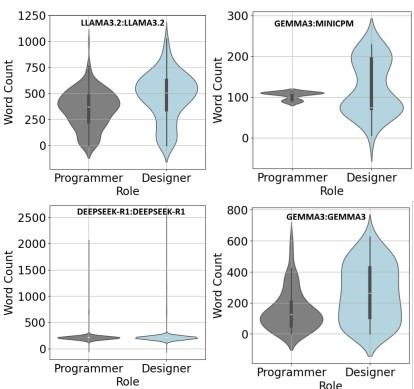

**Figure 7: Violin Plot of Designer vs Programmer Word Count**

In addition to the quantitative analysis so far, we also conducted a manual analysis of the conversations to check the role alignment in each agent pair and also which role was more responsible in initiating, diverging from and converging to a correct solution. In pairs like *Qwen3:Qwen3, Gemma3:MiniCPM, Gemma3:Gemma2, Llama 3.3:Llama 3.3, Gemma3: Deepseek-R1, and Llama 3.2:Llama 3.2*, the Designer generally talks of code refactoring & improved designs backed with partial or full implementation. Again, the Programmer fully implements the code in these pairs. For *Deepseek-R1: Deepseek-R1, Deepseek-R1:Llama 3.3, Qwen3:Deepseek-R1, and Deepseek-R1:MiniCPM*, both Designer and Programmer talk about Design Approach, Implementation, Testing, Analyzing, Validating, Explanation and Discussion. In *Gemma3:Gemma3* and *Llama 3.2: Deepseek- R1* it is observed that after few iterations the Designer and Programmer roles get swapped. In all the cases where the agent pairs initiate the required solution, it is the Programmer who leads the path. On the other hand, in most cases of divergence, it is observed that the designer initiates the same by talking about other related topics. Our manual analysis shows a strong alignment with the quantitative findings from BLEU, ROUGE, and word count distributions. Pairs with clearly separated roles also scored highly in semantic precision and recall, while pairs with mirroring or indistinct roles exhibited flat semantic metrics and symmetric verbosity. Additionally, cases of role reversal and delayed convergence in the manual analysis were reflected in rising BLEU/ROUGE trends over time. These correlations affirm the validity of our semantic metrics in capturing not only the outcome quality but also the evolving collaborative behavior between LLM agents.

**RQ4**: How does compilation success rate varies across agent pairs?

The compilation results are based on only C codes found as a part of agent conversation. The compilation outcomes across Designer:Programmer agent pairs are presented in Figure 8. We have not considered programs generated by the agents in other programming languages like C++, Python etc., for the purpose. Certain pairs like *Llama 3.2:Llama 3.2* and *Qwen3:Qwen3* demonstrate flawless performance with 100% compilation success and no observed failures or missing code, establishing them as the most reliable and effective configurations. In contrast, mid-tier performers such as *Qwen3:Deepseek-R1*, *Gemma3:Deepseek-R1*, and *Gemma3:MiniCPM* exhibit a more balanced distribution, with success rates ranging between 45-77%, moderate levels of compilation failure, and a non-trivial proportion of instances where no code is produced. More concerning are pairs like *Deepseek-R1:MiniCPM* and *Llama 3.3:Llama 3.3*, where success drops to nearly 30-40% and failure rates reach upto 62%, reflecting instability and a tendency to generate syntactically incorrect or incomplete code. Finally, pairs like *Gemma3:Gemma3*, and *Deepseek-R1:Deepseek-R1* outcomes 68-95% "No Code Found" cases. Since *Deepseek-R1:Deepseek-R1* was the only pair converging to the correct solution we further investigated the reason for the huge number of *No Code* cases. It was found that all these cases were associated with the Designer and Programmer both giving compilation and execution instructions. Overall, it was noticed that all the pairs with *Deepseek-R1* as one of the agents performed poorly in context of compilation. Though manual inspection revealed that the *No Code Found* cases were often related to compilation instructions (not random texts). Collectively, these results highlight the critical influence of model pairing on code quality, while some configurations consistently generate compilable code, others fail systematically, emphasizing the need for careful agent selection in automated coding pipelines.

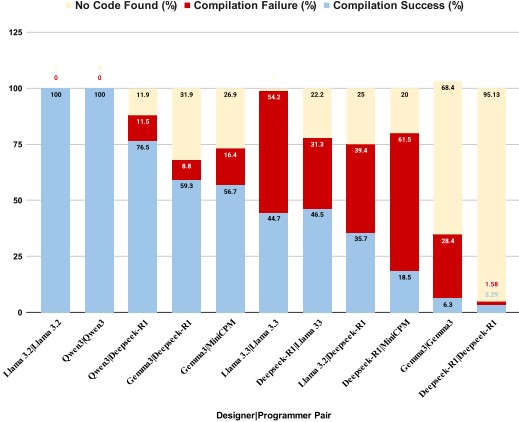

**Figure 8: Compilation Results**

Table 2 summarizes the nature of the conversation of three very diverse agent pairs. This cumulative picture shows that it is difficult to infer on the agent's capability based on a single factor. A good compilation success rate i.e., effectiveness could be accompanied with poor role alignment, again convergence can come along with a lot of semantic echoing. This is where our study is grounded and highlights the importance of investigating agent interaction patterns at a fine-grained level.

**Table 2: Results: Representative Summary**

| Agent Pair | Efficiency | Consistency | Effectiveness |
|---|---|---|---|
| DeepSeek-R1:DeepSeek-R1 | Convergence till last itr. | Semantic Echoing & Inconsistent Role Alignment | 3.29% |
| Qwen3:Qwen3 | Never started with the correct soln. | 37% Semantic Echoing & Strong Role Alignment | 100% |
| Llama3.3:Llama3.3 | Diverged after 69 itr. | No Semantic Echoing & Strong Role Alignment | 54.2% |

## 5 Discussion

Our study offers practical insight for the evolution from AI-assisted SE 3.0 [9], where LLMs serve as passive copilots to Autonomous SE systems and AI agents actively coordinate software development tasks. Our findings provide a blueprint for assembling specialized LLM agents suited for different stages of an autonomous SE pipeline.

i) **The Convergent Agents:** *DeepSeek-R1:DeepSeek-R1* was the only agent pair to converge correctly, but also exhibited strong semantic echoing and many "No Code Found" turns, which shows that successful convergence can coexist with interaction behaviors that would normally be considered problematic. ii) **Reasoning Paired with Non-Reasoning:** Pairs like *Gemma3:DeepSeek-R1* and *DeepSeek-R1:MiniCPM* illustrate the risk of semantic echoing without task progression. This suggests that reasoning models provide limited benefit when paired with non-reasoning models. While less successful in converging, such combinations highlight typical failure modes and can inform the design of health monitoring systems or robustness baselines for autonomous SE environments. iii) **Balanced Non-Reasoning Pairs:** Pairs of agents such as *LLaMA 3.2:LLaMA 3.2, LLaMA 3.3:LLaMA 3.3, and Qwen3:Qwen3* demonstrated promising collaborative behaviors. These same-model pairings were able to sustain role alignment and dialogic interaction, even when they did not fully converge to correct solutions. In practice, such configurations may be useful for tasks like requirement negotiation, design exploration, or refactoring, where longer and more balanced dialogues are beneficial. iv) **Compilation Friendly Pairs:** The differences across pairs indicate that model choice directly influences the produced quality of code. The top performing pairs were (*LLaMA 3.2:LLaMA 3.2* and *Qwen:Qwen* with 100% compilation success rate). This suggests that pair compatibility is an important factor when designing multi-agent SE systems. v) **Diverging Pairs & Termination/Stop Condition:** As seen in the results, the pairs *Qwen3:Deepseek-R1*, *Deepseek-R1:LLaMA 3.3*, and *LLaMA 3.3:LLaMA 3.3*, which eventually diverged from the correct solution after 3, 21, and 69 iterations respectively, had already produced the correct solution in the very first iteration. In other words, we observed no cases where agents began off-task and only later discovered the correct solution. This suggests that once a correct solution emerges, it tends to be preserved for a bounded number of iterations (≤ 69 in our study), while later recovery is unlikely. Behavioral signals extracted from the interaction trace, such as repetition rate, programming-topic adherence, role stability,

and related indicators may provide a basis for defining stopping conditions [7] in future multi-agent programming systems.

In multi-agent coding assistant systems, where one agent proposes a design and another implements it, a team may see that the agents continue producing compilable code while repeatedly revisiting the same ideas, drifting away from the requested task, or no longer responding meaningfully to each other. Our study provides the guidance on why and how the agent interaction traces should be investigated, rather than relying only on whether code is eventually produced. This is important because final code alone may hide whether the agents are still collaborating productively. In practice, monitoring the interaction trace can help detect early signs of failure, such as repetition, drift, or role instability before time and compute are wasted on unproductive exchanges.

## 6   Threats to Validity

We used the frameworks by Wohlin [25] and Staron [21] to structure our validity analysis. *Construct validity*: Measuring semantic alignment and solution quality through BLEU and ROUGE may not capture nuanced reasoning or non-literal correctness, especially when models paraphrase or restructure outputs. To mitigate this, we complemented metrics with manual annotations and convergence analysis. *Conclusion validity*: The observed behavioral patterns may not generalize across tasks or prompts, and programming languages since LLM performance is sensitive to input phrasing and sampling. Also, the Fibonacci game is open-ended and admits multiple reasonable implementations, which makes judgments of correctness and convergence sensitive to the specific evaluation criteria used. *External validity*: Our study is limited to 12 open-source LLM combinations with fixed prompts and role conditioning, which may not generalize to proprietary models or future versions. Real applications may involve more complex tasks, longer dialogues, and dynamic role shifts. Replication on broader model populations and tasks is needed to confirm our findings.

## 7   Conclusion and Future Work

attention to convergence behavior, role calibration, and termination policies.

This exploratory use-case study examined 12 *Designer:Programmer* LLM agent combinations to analyze multi-agent coordination in autonomous programming. We find that convergence, semantic alignment, and compilation success are not consistently correlated, strong alignment does not ensure correctness, and reasoning capability alone does not guarantee stable collaboration. Only *DeepSeek-R1:DeepSeek-R1* consistently converged to a correct, compilable solution, while several same-model pairs showed high alignment but failed to sustain correctness. Our results indicate that multi-agent performance depends more on role separation, pair compatibility, and interaction dynamics than on model size or lexical similarity alone. Notably, early solution emergence predicts eventual success, and divergence rarely recovers once initiated, highlighting the need for explicit stop conditions, monitoring mechanisms, and calibrated termination strategies in scalable multi-agent SE systems.

Future work will expand to more complex tasks, varied model pairings, and richer tool integration, while disentangling whether *DeepSeek-R1*'s convergence stems from reasoning design or model scale.

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

# 8  Acknowledgments

This paper has been partially financed by Software Center, www. software-center.se, a collaboration between Chalmers, the University of Gothenburg, and 17 companies. The paper has also been partially funded by the SFO Transport/AoA Transport at the University of Gothenburg and Chalmers University of Technology.

Received 2026-02-15; accepted 2026-03-28