# OpenReview forum: "Understanding Conversational Patterns in Multi-agent Programming: A Case Study on Fibonacci Game Development"
_ACM.org/AIWare/2026/Conference — AIware 2026_

### Official Review · Reviewer_f9Gu · 2026-03-07

**Rating:** 2
**Confidence:** 4

**Review:**

- How do you guarantee the generalizability of this problem domain? How can researchers or practitioners actually utilize these results in real software engineering workflows?
- How do you determine what the optimal convergence point is? The paper mentions convergence but does not clearly define what optimal convergence means or how it is detected.
- How do the authors measure semantic alignment? This is perhaps the hardest concept in the paper, but it is not clearly defined.
- BLEU score only checks word overlap. How do you guarantee that lexical similarity actually reflects semantic overlap?
- The statement “In a nutshell, this study establishes an initial empirical foundation for improving multi-agent LLM collaboration in programming contexts” seems overstated. There are already many papers on LLM-based agent collaboration, and the paper should acknowledge and cite them.
- The paper states “we employed computational experiment as a research method.” It is unclear what the relevance of “computational” is here. What distinguishes this from a standard experimental evaluation?
- Does orchestration help convergence, divergence, or drift? The current setup uses a very simple workflow. It is unclear whether the results would generalize when more complex orchestration is introduced (e.g., tool usage, planning, retrieval, or multi-step pipelines).
- The motivation behind the inductive topic analysis is unclear. What is the implication of these topics? For example, what if different topics still produce correct solutions?
- The paper briefly mentions the idea of early stopping, but does not clearly explain how the results inform early stopping strategies.
- The motivation and implication behind RQ3 are somewhat vague. For example, statements like “Designers were generally more verbose than Programmers, reflecting their role in specification and guidance” appear to be highly dependent on the specific prompts used in the experiment. Moreover, RQ3 attempts to analyze the “role” of agents, but the analysis mainly relies on BLEU scores and text similarity, which may not directly measure agent roles.
- The system design is essentially a self-repeating loop between two agents. Therefore, it is unclear whether the paper is truly measuring the “role” of agents. The observed behavior could simply be prompt drift or autoregressive stability, rather than collaborative problem solving. As a result, the paper reads more like an analysis of iterative LLM prompting stability rather than multi-agent collaboration.
- The statement “Overall, these results show that sustained semantic alignment depends more on clear role separation and interaction dynamics than on surface-level lexical overlap alone.” is difficult to interpret. It is unclear how the experiments support this conclusion.
- For RQ4, code can compile but still be completely incorrect. Therefore, how do the alignment or interaction metrics from RQ1 to RQ3 relate to actual solution correctness? For example, could a pair show high alignment but low correctness? The paper would benefit from a cross-level analysis that connects alignment metrics with functional correctness.

**Summary:**

This paper studies multi-agent collaboration between large language models in programming tasks. The authors design a two-agent system consisting of a Designer agent, which provides specifications, and a Programmer agent, which generates code based on those instructions. The interaction proceeds iteratively, where the output of one agent becomes the input to the other in a repeated loop. The paper then measures alignment, divergence and drift of this LLM interactions and check how these pattern correlate with agent capability to compile the code.

---

> ### Author Response · Authors · 2026-03-17
> **Thank you for the comments. The responses below follow the same sequence as the review comments.**
>
> 1. Scope: This paper is an initial case study of interaction patterns in a controlled multi-agent programming setting.
> Contribution: Its value is exploratory, identifying trace-level phenomena such as repetition, drift & role instability as signals for future validation. Practical implication: In a multi-agent coding assistant, agents may keep producing compilable code while drifting, repeating, or losing coordination. Our study suggests that these interaction traces should be monitored, since final code alone may hide unproductive collaboration. Such signals can support stopping, rerouting, or human review. We present the work an an initial empirical case study rather than giving broad general claims.
> 2. Successful Convergence = Agent pair ended with a correct (functionally correct, manually executed & verified their runtime behavior) + compilable solution.
>
> 3, 4 & 10. We use the term semantic alignment more operationally to refer to the extent to which consecutive Designer:Programmer responses appeared to remain aligned with each other across turns. This was approximated through BLEU/ROUGE-based scores and then supported with manual qualitative analysis of the conversations. We will clarify the concept more explicitly describing BLEU/ROUGE as indicators of response overlap rather than direct measures of alignment. The stronger interpretation of whether agents were collaborating with each other came from the accompanying manual analysis, not from the metric values alone.
>
> 5. We already discuss several emerging work in this area. Our intention was to position the paper as an initial empirical step within our specific line of inquiry, focusing on interaction dynamics & role-specialized collaboration in programming.  We will make the connection to prior work more explicit.
> 6.Computational experiment = Fully software-based + controlled experimental environment - human participants
> May be this distinction is not important in the paper. We will replace this using a more standard term like controlled empirical study (acc. to ACM SIGSOFT standards)
> 7.  Our aim was to keep the orchestration minimal so that we could observe the intrinsic interaction dynamics between Designer & Programmer agents with minimum intervention. The controller mainly provides role conditioning, turn management, & full conversational history, which helps maintain contextual continuity & supports basic role adherence, but does not actively guide the agents toward convergence through external planning or tool feedback. We will frame richer orchestration mechanisms as an important direction for future work (already exploring tool-augmentation).
> 8. We introduced the topic analysis to understand how agent attention moved over time during collaboration, whether the agents remained focused on the Fibonacci game task, explored related subproblems productively, or drifted into increasingly unrelated directions. In several cases, we observed that once the agents diverged from the original task, they did not return to solving the original problem. This made topic analysis important, not because any individual topic is “wrong,” but because it helped reveal when the agents were losing task focus.
> 9. We provided 100 iterations as a safe stopping point because for all the model conversations we analyzed, the agents started repeating after 100 iterations in almost all cases. But presenting it as a practical stopping point is stronger as we experimented with limited models. Our results show that non-convergent runs often exhibit patterns like sustained repetition, topic drift , & role instability, . Therefore, early stopping may be better guided by such interaction-level patterns derived from the trace, rather than by a simple fixed turn threshold.
> 11. We believe that the interaction in our study is more than a purely self-repeating loop. The controller explicitly maintained persistent role conditioning throughout the dialogue, when sending design-oriented outputs to the Programmer or implementation-oriented outputs to the Designer, it also preserved the corresponding role prompts in the interaction history, so that each agent continued to receive its role-specific context across turns. Our intent was therefore to study role-conditioned multi-agent interaction, rather than generic iterative prompting alone. Also, across multiple runs, we observed instances of Designer proposed solution directions, Programmer implemented or modified them, and the pair engaged in iterative refinement. Our setup represents a minimal form of role-conditioned multi-agent collaboration, not reflecting the full richness of collaborative SE.
> 12. In several cases, we observed model pairs with strong BLEU/ROUGE scores exhibited repetition. It was therefore cross-checked by manual analysis to ensure lexical analysis alone does not decides semantic alignment.
> 13.  In the revision, we will modify Table 2 to include a clear cross-level interpretation (alignment with correctness) .

---

> > ### Comment · Reviewer_f9Gu · 2026-03-18
> >
> > Thank you to the authors for the detailed responses and revisions. I am satisfied with the clarification,
> > and the addition of item (11) in the revision.

---

> > > ### Author Response · Authors · 2026-03-19
> > >
> > > Thank you. We value your feedback.

---

### Official Review · Reviewer_QJJA · 2026-03-11

**Rating:** 3
**Confidence:** 4

**Review:**

Strengths:
+ The paper addresses a timely and important question: how role-based LLM agents behave over sustained interaction, not just on final outputs.
+ The experimental setup is simple in a good way. The two-role design, fixed prompts, and controller-based logging make the study easy to understand and reproduce.
+ The paper looks at the conversations from several angles, including qualitative patterns, topical drift, lexical overlap, word counts, and compilation outcomes.
+ The paper provides prompts, infrastructure, compiler information, and a replication package.

Weaknesses:
- The paper relies heavily on the idea of a "correct solution", but that notion is not fully clear for an open-ended task that may allow multiple valid designs.
- It is not clear whether each model pair was run once or multiple times, which matters because the setup uses stochastic decoding.
- Several factors are mixed together in the pairing design, including model family, size, symmetry, and reasoning specialization, which makes some interpretations harder to support.
- BLEU and ROUGE are useful as surface overlap measures, but they do not always capture productive collaboration, especially when echoing is present.
- Some discussion claims, especially about reasoning-model necessity and stop conditions, are a bit stronger than the current evidence seems to support.

Detailed Comments:

- The paper would benefit from a clearer definition of correctness, convergence, and divergence. Since the task is open-ended and may admit more than one reasonable implementation, readers need to know how correctness was judged and how borderline cases were handled.

- It is also not clear whether each model pair was run once or multiple times. Because the controller uses temperature 0.5 and several conclusions depend on trace-specific behavior, some discussion of run-to-run variability would help readers judge how stable these patterns are.

- The pair design gives useful coverage, but it also combines several factors at once: family, size, symmetry, and reasoning specialization. For that reason, I would suggest keeping the interpretation mainly descriptive rather than causal.

- BLEU and ROUGE are reasonable as partial indicators of overlap between roles, but the paper itself shows that very high scores can reflect copying or echoing rather than healthy collaboration. It would help to frame these metrics as limited indicators, or to connect them more directly to the manual role-analysis results.

- The compilation analysis would be easier to interpret if it separated "no C code found" from "code was produced, but not in C". The 60% cutoff used in Table 2 for effectiveness also needs a short justification.

- One useful message of the paper is that correctness, lexical alignment, and compilability do not necessarily move together. That is relevant for researchers and practitioners building multi-agent software engineering systems.

- At the same time, the practical recommendations in the discussion should be framed a bit more carefully. Claims about needing at least one reasoning model or using roughly 100 turns as a stop condition feel somewhat stronger than the current design can fully support.

- I would encourage the authors to separate more clearly what this study shows from what it suggests as a direction for future testing.

- The novelty of the paper is mainly empirical. The value lies in the close look at interaction dynamics and failure modes, not in a new framework or metric. That would be a fair and worthwhile contribution.

- The paper could make their contribution more clear by explaining what the proposed pattern categories add beyond prior work on multi-agent failure modes or agent reasoning traces.

- The paper would still benefit from bringing a few important details into the main text rather than leaving them only in the artifact, especially the correctness rubric, code extraction rules, topic taxonomy summary, and model snapshot details.

- The threats-to-validity section is a good start, but it could more directly discuss the open-ended nature of the task, the possible effect of not repeating runs, and the role of manual judgment in the annotations.

- Table 2 is useful as a summary, but the yes/no categories may compress the results more than necessary. A slightly more nuanced summary might better match the earlier analysis.

**Summary:**

This paper examines two-agent LLM collaboration for programming in a simple Designer/Programmer setting managed by a Python controller. The agents work on one open-ended task, "Write a mathematical game of Fibonacci". The study compares 12 model pairings from several open-source LLM families and looks at three dimensions: efficiency (whether pairs reach and keep a correct solution), consistency (BLEU/ROUGE, word counts, and manual role analysis), and effectiveness (compilation success). The main reported result is that DeepSeek-R1:DeepSeek-R1 is the only pair that stays on a correct solution throughout the interaction, while several other pairs either drift away or show strong lexical alignment without correctness. The goal of the paper is to understand interaction dynamics and failure modes rather than propose a new multi-agent framework.

I found the paper interesting and worth developing further. Studying process-level agent behavior, rather than only final task success, is valuable, and the paper brings out several useful observations about echoing, drift, shallow interaction, and the loose connection between alignment, compilation, and correctness. The setup is also clear, and the mix of qualitative and quantitative analysis makes the study easy to follow.

My main reservation is about how far the current design can support the broader claims in the discussion. As written, the paper reads best as an exploratory case study built around one open-ended task and a limited set of model pairings. In that setting, the observations are useful, but some claims would benefit from more careful phrasing and a little more methodological detail. The paper would be stronger if it clarified how "correct solution" was judged, explained whether the reported traces are stable across reruns, and presented the alignment and compilation results a bit more cautiously.

---

> ### Author Response · Authors · 2026-03-17
>
> **Thank you for the thoughtful observations. The responses below follow the same sequence as the review comments.**
>
>  1. In our evaluation, correctness was judged as a combination of compilation and manual evaluation (execution). A solution was treated as correct if it both compiled and produced functionally appropriate behavior at runtime. We can provide a good definition and explanation in Section 3.3 of the revised work.
>
>  2. In our setup, the temperature was set to 0.5 after preliminary experiment with 0.3, 0.7, and 1.0 (less stability & more hallucination). The experiments were not based on a single trace only but each model-pair configuration was run twice under the same experimental setup to check whether the interaction patterns differed substantially across runs. In these repeated runs, we did not observe major differences that changed the overall end results reported in the paper. For space scarcity, we did not discuss the repeated runs in the paper. In the revision we will add a brief discussion of run-to-run variability as both a stability check in the current study and a direction for more systematic future analysis.
>
> 3. We will revise the Discussion, particularly to emphasize interaction-level observations across model families.
>
>  4. In the paper, our intention was to use  the BLEU/ROUGE metrics as lightweight signals of response overlap across consecutive Designer:Programmer turns backed with manual analysis. As the reviewer correctly notes, very high scores can sometimes reflect copying, semantic echoing, or shallow repetition rather than productive collaboration. This is in fact consistent with our own qualitative findings. In the revision, we will clarify that BLEU/ROUGE are used as exploratory overlap indicators, and that interpretations about role alignment are based on their combination with the manual qualitative analysis rather than on the metric values alone.
>
> 5. Regarding the 60% cutoff in Table 2, our intention was to use it as a practical descriptive threshold for summarizing whether a model pair produced compilable C code in a majority of attempts. In the revision, we will describe 60% as an operational threshold used for coarse comparative categorization and also modify Figure 8 to contain both C and Non-C codes (we already have the annotated data for it).
>
> 6. We will revise the Discussion section to emphasize the decoupling across the notions of  correctness, alignment, & compilability.
>
> 7. We will revise the discussion to soften the current wording around “reasoning models,” since our design was not intended to provide a strong claim about any particular model type. We had provided 100 iterations as a safe stopping point because for all the model conversations we analyzed, the agents started repeating after 100 iterations in almost all cases. We also agree that presenting 100 turns as a practical stopping point is stronger as we experimented with limited models. In the revision, we will replace this with a more cautious discussion suggesting that behavioral signals extracted from the interaction trace, such as repetition rate, programming-topic adherence, role stability, etc. may provide a basis for defining stopping conditions in future multi-agent programming systems.
>
> 8. In the revision, we will explicitly restructure the Discussion section around two levels of interpretation:
> i) Study-supported findings, limited to what was directly observed in our data and analysis & ii)Hypotheses and Future directions, covering ideas that are motivated by observations but not conclusively established by the current design (e.g.,  model usefulness, stopping-condition design, and improved alignment metrics).
>
> 9. We will sharpen this positioning further and ensure that the novelty is framed more clearly around the observational and process-level insights of the study.
>
> 10. We agree that the contribution of the proposed pattern categories in comparison to existing works should be clarified more explicitly. We will make this distinction clear in the discussion section of the revised version.
>
> 11. In the revision, we will include a clearer description of the correctness rubric and the code extraction rules, since these are central to understanding how outcomes were evaluated. For the topic taxonomy, the main topic groups are already summarized in Figure 4. Since the full taxonomy is large, we provided it in the replication package. We will make this more explicit in the paper.
>
> 12. We will revise the threats to validity section to discuss about the open-ended nature of the Fibonacci game task, stability of trace-specific behaviors under stochastic decoding & subjective interpretation due to manual analysis.
>
> 13. Table 2 originally contained more detailed information, but we simplified it to a binary summary for space reasons. In the revision, we will try to make this table more informative by incorporating a more subtle summary of the underlying observations that led to the Yes/No labels.

---

> > ### Comment · Reviewer_QJJA · 2026-03-18
> >
> > I thank the authors for answering the comments. The responses to Comments 1, 4, 5, 6, 7, 8, 12, and 13 are generally sufficient and reasonable, while Comments 2, 3, 9, 10, and 11 are related to subsequent revisions. Overall, I am satisfied with the responses.

---

> > > ### Author Response · Authors · 2026-03-18
> > >
> > > Thank you again for your insightful comments. We will do the required revisions against comments 2, 3, 9, 10, and 11 in the camera ready version if the paper is accepted.

---

> > > ### Author Response · Authors · 2026-03-18
> > >
> > > Thank you. We appreciate the feedback.

---

### Official Review · Reviewer_qNfG · 2026-03-13

**Rating:** 2
**Confidence:** 3

**Review:**

Strengths:
+ Relevant and Timely Topic


Weaknesses

- The paper defines correctness entirely in terms of whether the generated C code compiles under GCC. Compilation is a syntactic check, not a semantic one. A program that compiles may still produce incorrect sequences, crash at runtime, fail to function as a game, or produce no meaningful output. None of these failure modes is detected by the evaluation.

- The findings reduce to one takeaway. A reasoning model paired with itself produced compilable code more reliably than other combinations. This is neither surprising. The field already understands that reasoning-capable models tend to outperform general-purpose ones on structured tasks.

- BLEU and ROUGE were designed for natural language evaluation and have known limitations when applied to code. Two functionally equivalent programs may share very few surface tokens, while two syntactically similar programs may differ substantially in behavior. Using these metrics to measure role alignment in a code generation task needs stronger justification or supplementation with code-aware measures such as CodeBLEU.

- The paper alternates between "ROGUE" and "ROUGE" in several sections including Section 3.3. This should be corrected consistently throughout the manuscript.

**Summary:**

This paper reports an exploratory empirical study of designer-programmer agent pairs, each instantiated from one of seven open-source LLMs, tasked with writing a Fibonacci-based game in C. The study examines different dimensions of agent behavior: convergence to a correct solution, alignment between agents as measured by BLEU and ROUGE, and compilation success rate. The central finding is that the DeepSeek-R1:DeepSeek-R1 pairing was the combination that produced and consistently maintained a correct, compilable solution.

---

> ### Author Response · Authors · 2026-03-17
> **Clarifying i) correctness beyond compilation, ii) contribution, iii) BLEU/ROUGE for role alignment & iv) ROUGE/ROGUE inconsistency**
>
> Thank you for the thoughtful observations. We tried to respond in the best possible way and we hope that this makes the study more interesting and interpretable to the readers
>
> ### 1. Correctness beyond compilation
>
> **Reviewer comment:** The paper defines correctness entirely in terms of whether the generated C code compiles under GCC. Compilation is a syntactic check, not a semantic one. A program that compiles may still produce incorrect sequences, crash at runtime, fail to function as a game, or produce no meaningful output. None of these failure modes is detected by the evaluation.
>
> **Response:** We agree that compilation alone is only a syntactic check and does not establish semantic or functional correctness. However, our evaluation was not based solely on GCC compilation, but also on expert evaluation. In the paper, we defined Successful Convergence as the case where an agent pair ultimately arrived at a correct, compilable solution. By “correct,” we meant functionally correct and we manually executed the generated programs and verified their runtime behavior against the expected task outcome, rather than relying only on successful compilation.
>
> **Action:** We will have a clearer definition of solution convergence and also make the manual evaluation process clear in Section 3.3.
>
> ### 2. Main takeaway and broader contribution
>
> **Reviewer comment:** The findings reduce to one takeaway. A reasoning model paired with itself produced compilable code more reliably than other combinations. This is neither surprising. The field already understands that reasoning-capable models tend to outperform general-purpose ones on structured tasks.
>
> **Response:** We agree that, taken literarily, the finding that a reasoning-capable model pair was the only pair to consistently converge to a correct solution may not by itself be surprising. However, our intended contribution was interaction dynamics in role-based multi-agent programming, with four research questions covering conversational patterns, topic drift, role alignment, and compilation behavior, rather than a simple model-ranking exercise.
>
> In addition, the study highlights several other findings: 1) some pairs began with a correct solution but later diverged permanently, 2) many pairs entered echoing or shallow interaction modes, and 3) different model combinations exhibited distinct stability profiles over long conversations. These observations support our broader claim that the key challenge in multi-agent programming is not only raw model capability, but also how role-specialized agents coordinate, remain aligned, and avoid breakdown over time. This matters because real-world multi-agent programming depends on sustained collaboration quality, not just isolated model strength; understanding where convergence, alignment, and compilability come apart is essential for designing reliable agentic workflows.
>
> **Action:** We can emphasize these points in the discussion section even more.
>
> ### 3. BLEU/ROUGE for role alignment
>
> **Reviewer comment:** BLEU and ROUGE were designed for natural language evaluation and have known limitations when applied to code. Two functionally equivalent programs may share very few surface tokens, while two syntactically similar programs may differ substantially in behavior. Using these metrics to measure role alignment in a code generation task needs stronger justification or supplementation with code-aware measures such as CodeBLEU.
>
> **Response:** In our work, we use BLEU and ROUGE metrics as exploratory indicators of response similarity between consecutive Designer:Programmer turns in a mixed conversational setting, where responses often contained varying proportions of natural language, partial/full code, and explanatory text. We considered CodeBLUE at the beginning, but because many adjacent turn pairs were not pure code-to-code comparisons, code-specific metrics such as CodeBLEU could not be uniformly applicable across the full dialogue trace. We therefore chose BLEU and ROUGE followed with manual analysis to determine role alignment.
>
> **Action:** We will clarify this motivation in the revision and explicitly discuss this choice as a limitation.
>
> ### 4. ROUGE/ROGUE inconsistency
>
> **Reviewer comment:** The paper alternates between "ROGUE" and "ROUGE" in several sections including Section 3.3. This should be corrected consistently throughout the manuscript.
>
> **Response:** We acknowledge the inconsistency in the use of “ROGUE” and “ROUGE” in the manuscript, including Section 3.3. This is a typographical error, and we will carefully revise the paper to ensure that the metric name is written consistently as ROUGE throughout.